

# Non-visual exploration of novel objects increases the levels of plasticity factors in the rat primary visual cortex

Catia M. Pereira[1,*], Marco Aurelio M. Freire[2,*], José R. Santos[3], Joanilson S. Guimarães[4], Gabriella Dias-Florencio[5], Sharlene Santos[5], Antonio Pereira[6] and Sidarta Ribeiro[5]

[1] Instituto Internacional de Neurociências de Natal Edmond e Lily Safra, Macaiba, RN, Brasil

[2] Programa de Pós-graduação em Saúde e Sociedade, Universidade do Estado do Rio Grande do Norte, Mossoró, RN, Brasil

[3] Departamento de Biociências, Universidade Federal de Sergipe, Itabaiana, SE, Brasil

[4] Instituto de Ciências Biológicas, Universidade Federal do Pará, Belém, PA, Brasil

[5] Instituto do Cérebro, Universidade Federal do Rio Grande do Norte, Natal, RN, Brasil

[6] Faculdade de Engenharia Elétrica, Universidade Federal do Pará, Belém, PA, Brasil

[*] These authors contributed equally to this work.

Corresponding author
Sidarta Ribeiro,
sidartaribeiro@neuro.ufrn.br

## ABSTRACT

**Background**. Historically, the primary sensory areas of the cerebral cortex have been exclusively associated with the processing of a single sensory modality. Yet the presence of tactile responses in the primary visual (V1) cortex has challenged this view, leading to the notion that primary sensory areas engage in cross-modal processing, and that the associated circuitry is modifiable by such activity. To explore this notion, here we assessed whether the exploration of novel objects in the dark induces the activation of plasticity markers in the V1 cortex of rats.

**Methods**. Adult rats were allowed to freely explore for 20 min a completely dark box with four novel objects of different shapes and textures. Animals were euthanized either 1 ($n = 5$) or 3 h ($n = 5$) after exploration. A control group ($n = 5$) was placed for 20 min in the same environment, but without the objects. Frontal sections of the brains were submitted to immunohistochemistry to measure protein levels of egr-1 and c-fos, and phosphorylated calcium-dependent kinase (pCaKMII) in V1 cortex.

**Results**. The amount of neurons labeled with monoclonal antibodies against c-fos, egr-1 or pCaKMII increased significantly in V1 cortex after one hour of exploration in the dark. Three hours after exploration, the number of labeled neurons decreased to basal levels.

**Conclusions**. Our results suggest that non-visual exploration induces the activation of immediate-early genes in V1 cortex, which is suggestive of cross-modal processing in this area. Besides, the increase in the number of neurons labeled with pCaKMII may signal a condition promoting synaptic plasticity.

## INTRODUCTION

At the beginning of the last century, Korbinian Brodmann proposed that the neocortex is divided into several cytoarchitectonically distinct areas, each one devoted to a distinct function (*Brodmann, 2006*; *Guimaraes, Santos & Freire, 2016*). This classical finding was subsequently reinforced by several studies using multiple experimental techniques, including tract-tracing studies showing that thalamocortical inputs from a given sensory modality preferentially target specific primary cortical areas (*Hubel & Wiesel, 1959*; *Mountcastle, 1997*; *Pereira et al., 2000*; *Rocha et al., 2007*; *Santiago et al., 2007*; *Dooley et al., 2015*). However, a growing number of recent studies have shown that neuronal responses in those areas are characterized by a large amount of sensory crosstalk (*Lemus et al., 2010*; *Vincis & Fontanini, 2016*). For instance, whisker stimulation in rats generates patterns of subthreshold cortical activation that spread far from the traditional limits of the primary somatosensory cortex (*Frostig et al., 2008*). This pattern of activation means that unimodal sensory stimulation can potentially modulate the activity of neurons located in sensory areas related to other sensory modalities, despite the existence of cytoarchitectural borders (*Stehberg, Dang & Frostig, 2014*).

It remains to be determined, however, whether such cross-modal interactions would impact beyond the transitory modulation of neuronal activity, to trigger longer-lasting plastic changes in primary sensory cortices. Monocular enucleation during adulthood elicits cross-modal reorganization of the visual cortex, a phenomenon supported by preexisting projections from the somatosensory cortex (*Van Brussel, Gerits & Arckens, 2011*). Thus, cross-modal plasticity may help individuals recover from sensory deprivation due to injury of the neural pathways associated with a specific modality (*Rabinowitch & Bai, 2016*).

In a previous study, we showed that the whisker-based exploration of novel objects in the darkness leads to a marked upregulation of the mRNA levels of calcium-dependent immediate-early genes (IEGs) in primary visual (V1) cortex, which are associated with memory consolidation (*Ribeiro et al., 2007*). Using multielectrode recordings and the same free-exploration paradigm, we also showed in a later study that 35% of the V1 neurons significantly change their firing rates during whisker-based object exploration, despite the lack of visual inputs (*Vasconcelos et al., 2011*). Most importantly, neuronal ensembles in V1 cortex were shown to carry non-visual information useful to discriminate among different objects (*Vasconcelos et al., 2011*).

As a followup to these earlier studies, in the present work we sought to contribute with additional information concerning the modulation of visual cortex induced by tactile stimulation. We investigated the causes and consequences of cross-modal mRNA IEG induction in V1 cortex, by assessing the levels of phosphorylated CaMKII, a calcium-dependent kinase upstream of *c-fos* and *egr-1* transcripts with a well-documented role in plasticity (*Kaczmarek & Chaudhuri, 1997*). To examine the response downstream, we assessed the levels of c-fos and egr-1 proteins (*Bading, Ginty & Greenberg, 1993*; *Kaczmarek, 2000*; *Jones et al., 2001*; *Nunes et al., 2010*).

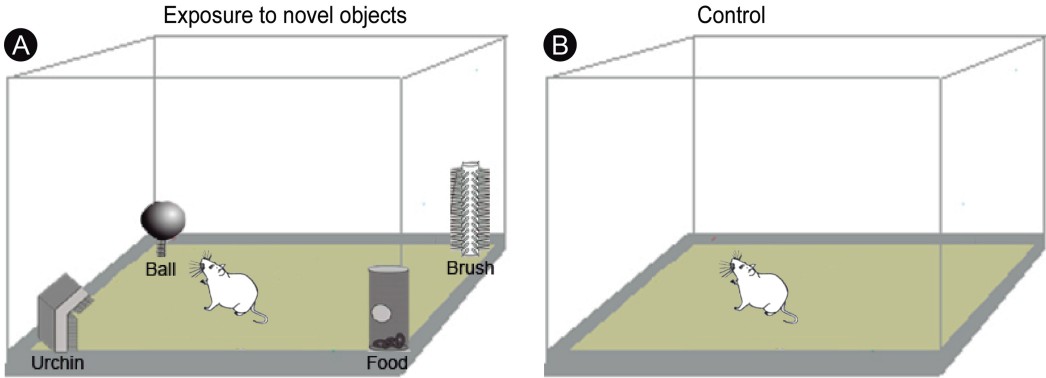

**Figure 1  Experimental design.** (A) Experimental animals were transferred to a box and allowed to freely explore four novel objects of different shapes and textures ('ball', 'urchin', 'brush' and 'food') in the dark during 20 min, being returned to their regular cages after the end of exploration. (B) Control animals were placed for 20 min in the same box, but without the objects. Drawing adapted from *Ribeiro et al. (2007)*.

## MATERIALS AND METHODS

### Animals

Fifteen adult male Wistar rats (5 months-old, $320 \pm 20$ g) were used in this research. All experimental procedures were approved by the Institutional Local Ethics Committee for the Use of Animals (ID 05/2007), in accordance with the NIH Guidelines for the Care and Use of Laboratory Animals. All efforts were made to avoid suffering and to reduce the number of animals used. Prior to the experiment, all animals remained in conventional cages ($49 \times 34 \times 16$ cm) under standard conditions (lights on at 07:00, lights off at 19:00, $22 \pm 2$ °C) with free access to food and water.

### Object exploration in the dark

All experimental procedures began at 22:00 under infrared light, to ensure that the animals were kept unstimulated by visible light for 3 h. A group of animals ($n = 10$) was transferred to a closed box ($55 \times 40 \times 20$ cm) and allowed to freely explore four novel objects of different shapes and textures (termed 'ball', 'urchin', 'brush' and 'food') for 20 min (Fig. 1A). The animals were then returned to their home cages and euthanized after one ($n = 5$) or three ($n = 5$) hours after exploration. A control group of animals ($n = 5$) was placed for 20 min in the same box in the dark, but without the objects (Fig. 1B), returned to their home cages and then euthanized after 1 h. Home cages were free of objects and the animals were well adapted to them, which precluded that prior environmental stimuli could induce IEG activation. To evaluate the degree of locomotion of the animals during the post-exploratory period, the total distance traveled and the mean velocity were video recorded and quantified using Any-maze software (version 4.3; Stoelting Co., Wood Dale, IL, USA).

### Immunohistochemistry

All animals were deeply anaesthetized with sodium pentobarbital (90 mg/kg, i.p.) and decapitated. The brains were quickly removed and fast-frozen in an embedding medium (Tissue Tek; Sakura Finetek, Tokyo, Japan). Then, the brains were frontally sectioned at

20 μm in a cryostat (Carl Zeiss Micron HM 550; Carl Zeiss AG, Oberkochen, Germany) and thaw-mounted over electrically charged glass slides following a serial distribution. Brain slices were then post-fixed for 15 min in 4% paraformaldehyde in 0.1 M phosphate buffer (PB). For immunohistochemistry, sections were washed during 20 min in 0.1 M phosphate buffer-Tween (PB-T) and incubated in a blocking buffer (BB) solution (0.5% fresh skim milk and 0.3% Triton X-100 in 0.1M PB) for 30 min. Sections were incubated overnight at 18 °C in primary antibody (1:200 in BB; phosphorylated polyclonal rabbit CaMKII; Millipore Corp., Burlington, MA, USA, catalog #AB3827; rabbit polyclonal egr-1 and c-fos, Santa Cruz Biotechnology, Inc., Santa Cruz, CA, USA, catalog #sc-189 and # sc-52, respectively), washed in PB-T (2×, 10 min each), incubated with a biotinylated secondary goat anti-rabbit antibody (1:200 in BB; Vector Labs, Burlingame, CA, USA) for 2 h, washed in PB-T (2×, 10 min each), and incubated in avidin-biotin-peroxidase solution (Vectastain Standard ABC kit; Vector Labs, Burlingame, CA, USA) for 2 h. Next, slides were placed in a solution containing 0.03% 3,3′ diaminobenzidine (DAB) and 0.001% hydrogen peroxide in 0.1 M PB (*Freire et al., 2005*). Primary antibodies were replaced by normal serum in some test sections. The reaction was monitored every 30 min and was interrupted by washing sections in 0.1 M PB during 5 min. In order to avoid artifacts due to procedural inconsistencies, all sections of all groups were processed together, incubated simultaneously and spent an identical time in the same DAB solution. Alternate sections were stained with cresyl violet (Nissl staining) to reveal cyto-architectonic boundaries. At the end of all procedures, sections were dehydrated and cover-slipped with Entellan (Merck, Darmstadt, Germany). Digital images were acquired with a CX9000 camera (MBF Bioscience Inc., Williston, VT, USA) attached to a Nikon Eclipse 80i optical microscope (Tokyo, Japan—4×, 10× and 20×objectives) under the same light parameters, to avoid bias. The contrast, and/or brightness of pictures were adjusted using Photoshop CS5 software (Adobe Systems Inc., San José, CA, USA).

## Data quantification

IEG-labeled cells (three sections/animal, $n = 5$ animals per group for every marker) were counted with the *Neurolucida* system (MBF Bioscience Inc., Burlington, VT, USA) using 10 grids of $100 \times 100$ μm for each section across V1 cortex. Analysis of pCaMKII staining in V1 cortex was done by optical densitometry with the ImageJ software (http://rsb.info.nih.gov/ij/) using a 0.05 mm$^2$ square window ($n = 5$ animals per group, three sections per animal; five samples per section). The low reactivity in the white matter was used as background reference (signal was averaged over five different sites using the same square window). For each animal, the average optical density (OD) was named $G$, cortical white matter $W$ and a contrast index $C$ was calculated according to the equation: $C = (G - W)/(G + W)$ (*Freire et al., 2007*). All photographs were taken with the same illumination and microscope (Nikon Eclipse 80i; Nikon, Tokyo, Japan). Average values for all measurements were normalized to the levels observed in the control group.

## Statistics

Significant differences across groups were assessed with the non-parametric Kruskal Wallis test, with Dunn *post-hoc* test ($\alpha = 0.05$), using the Graphpad Prism 5.0 software

(GraphPad Software Inc., La Jolla, CA, USA). The numerical values represented in boxplots were presented as median with indication of the 25th to 75th percentiles.

## RESULTS

To ensure that the analyses involved only the region of interest associated with V1 cortex, its boundaries were defined by Nissl staining (Fig. 2A). In a qualitative analysis, immunohistochemistry with specific antibodies revealed a robust increase of egr-1 and c-fos labeling in V1 cortex of animals killed 1 h after object exploration in darkness, with a subsequent decrease after 3 h (Fig. 2B). However, the pattern of activation at the 3 h time period was still higher than the basal levels found in unexposed controls (Fig. 2B). The levels of phosphorylated CaMKII in V1 cortex followed a similar time course (Fig. 2B).

Although we had not quantified the intensity of immunolabeling in c-fos and egr-1-reactive cells, the general pattern of reactivity allowed us to identify cell profiles varying from intensely to weakly reactive in all evaluated time points. There was a marked prevalence of intensely reactive cells 1 h after exploration, a pattern markedly different from that seen at 3 h, which was more balanced (Fig. 2B). The effectiveness of the immunolabeling method for all markers could be demonstrated by the absence of labeling when the primary antibody was replaced by normal serum (test sections). Since the brains were collected fresh and immediately fast-frozen, some blood normally remains in the tissue. This likely explains the differences in background labeling observed across animals in the present study.

A quantitatively analysis confirmed that the peak of activity in V1 cortex occurred after 1 h of exploration in darkness for all markers (Fig. 3) (Kruskal–Wallis, Dunn *post-hoc* test, $^*p < 0.05$). After 3 h of exploration the expression levels for all markers in V1 cortex were lower when compared to those seen at 1 h of exploration, being similar to control levels (Fig. 3). Locomotion parameters (total traveled distance and mean velocity) measured during the post-exploration period revealed a similar behavioral pattern in all groups (Fig. 4A), with the animals spending equivalent amounts of time moving in the home cage (Fig. 4B).

## DISCUSSION

In the present work, we investigated whether the whisker-based exploration of novel objects in the dark is capable of increasing the protein levels of molecular markers of plasticity in the V1 cortex. In a previous study using the same experimental paradigm, we have demonstrated that it increases the mRNA levels of calcium-dependent IEGs in the V1 cortex (*Ribeiro et al., 2007*). Here we provide evidence that such non-visual exploration also increases the levels of phosphorylated CaMKII, as well as c-fos and egr-1 proteins. This suggests that the cross-modal response in the V1 cortex can lead to long-term plasticity.

In a recent study in rats, Nakajima and coworkers showed that visual deprivation induces a sustained increase of neural activity in the S1 cortex (*Nakajima et al., 2016*). The up-regulation of neuronal responsivity in the S1 cortex after visual deprivation can be explained by an strengthening of synaptic activity mediated by NMDA and/or AMPA receptors, which induces IEG expression (*Kaczmarek & Chaudhuri, 1997*). Our present

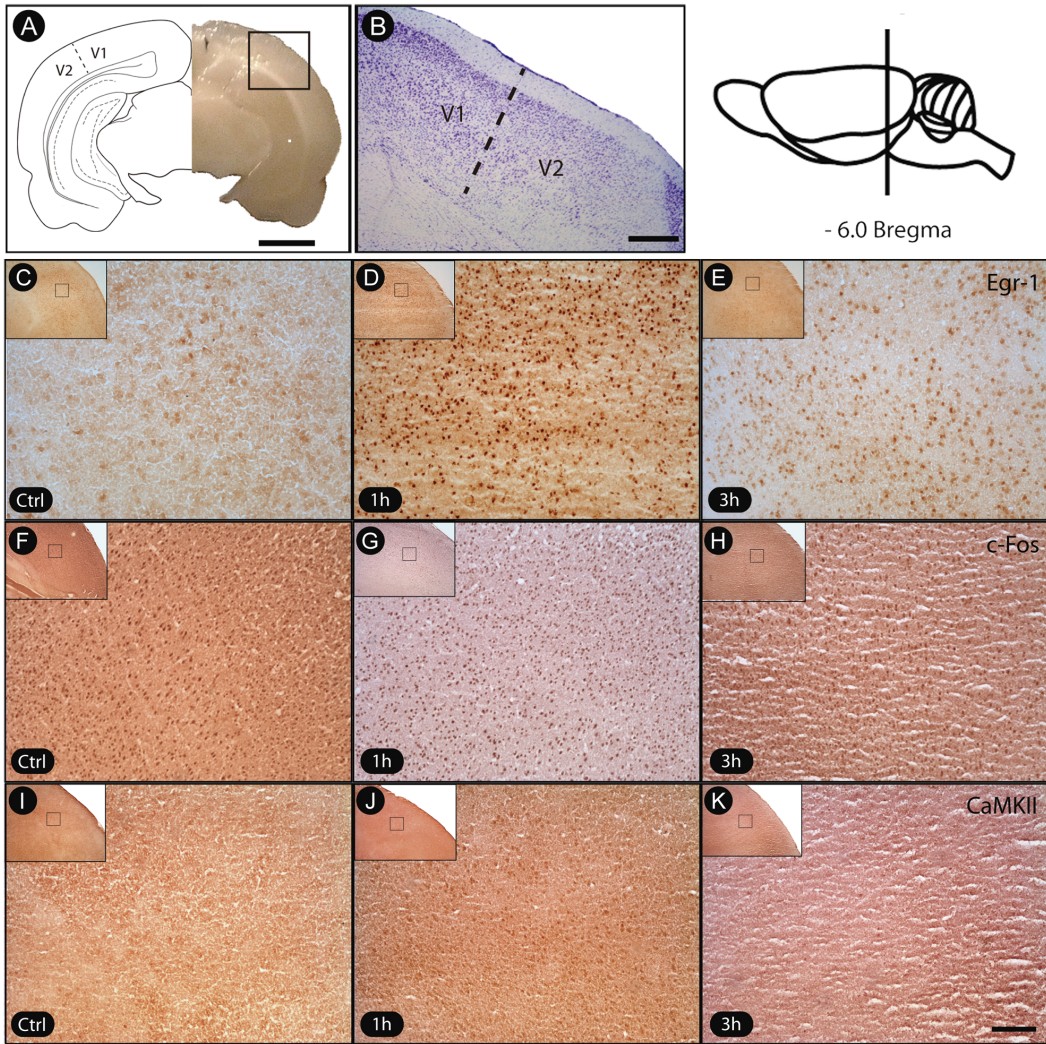

**Figure 2** **Transient activation of plasticity-related factors in V1 cortex after non-visual object exploration.** (A) Drawing indicating the location and border V1/V2 and the general aspect of the brain, immediately prior to sectioning. (B) Nissl-stained frontal section showing V1 cortex, obtained from the region indicated with the black square in A. The dashed line indicates the border between V1 and V2. (C–K) Sections immunostained for egr-1, c-fos and pCaMKII. Controls were unexposed to the objects, and experimental animals were assessed 1 h and 3 h after new object exposure. Notice the increased labeling after 1 h of exposure for all markers. Black squares on low power pictures indicate location from where the enlarged pictures were obtained. Scale bars: 3 mm (A); 200 µm (B); 100 µm (C–K).

results showing IEG activation in V1 cortex after non-visual exploration in the dark are supported by a recent report of long-term spatiotemporal changes in *c-fos* and *arc* levels in this area after bilateral deafness in adult rats (*Pernia et al., 2017*). According to this study, the disruption of the auditory input induces thalamocortical reorganization with consequent modification in both auditory and visual cortical circuits, which leads to the overexpression of IEGs in V1 cortex and a further re-expression of these markers in the

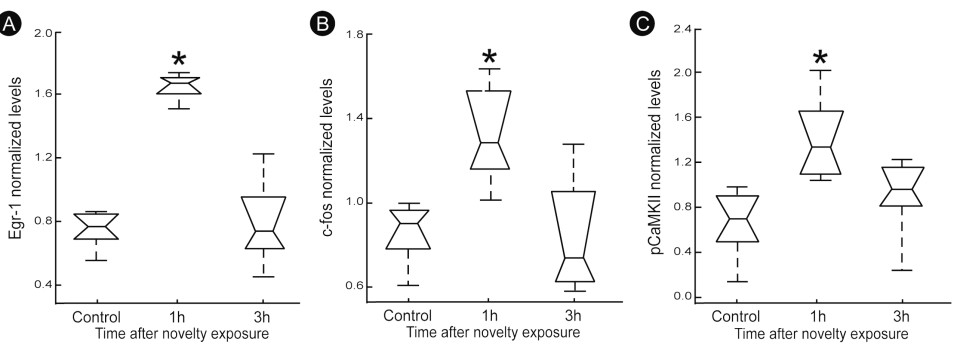

**Figure 3  Normalized levels plasticity-related factors in V1 cortex after non-visual object exploration.**
Group data for (A) egr-1, (B) c-fos and (C) pCaMKII, normalized in each panel by the mean value across groups. There was a significant increase 1 h after object exploration in V1 cortex for all markers ($n = 5$ animals per group, median $\pm$ quartiles of values for all brain sections assessed, Kruskal–Wallis, Dunn *post hoc* test, $^*p < 0.05$ corrected for the number of comparisons).

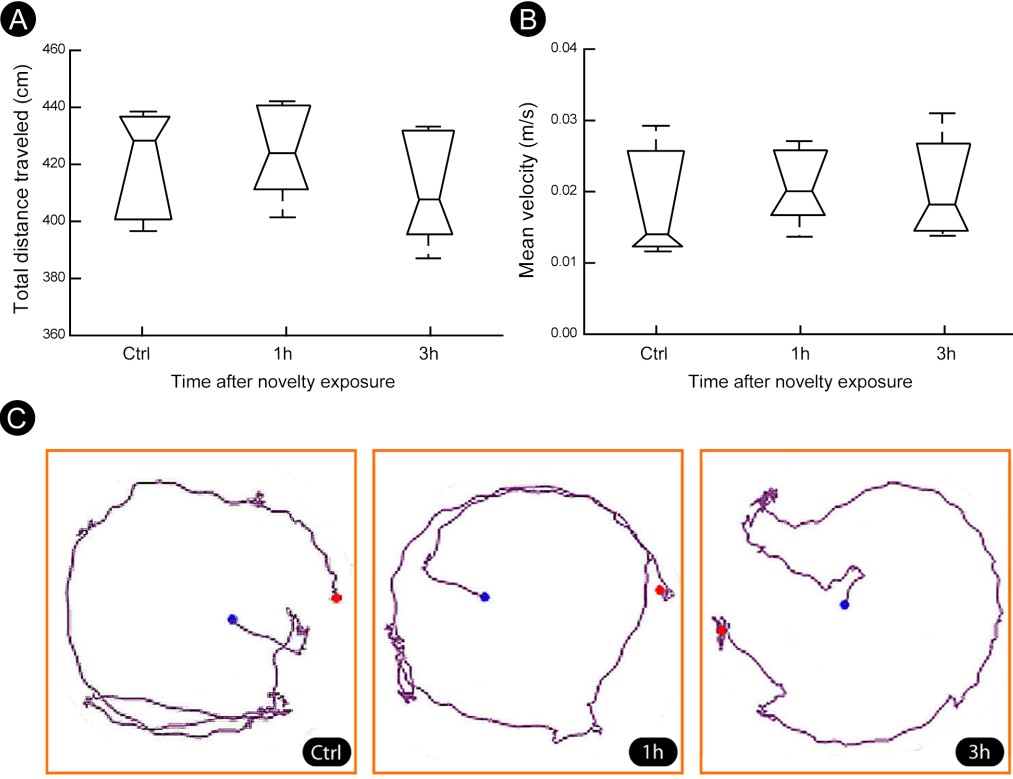

**Figure 4  Locomotion parameters during the post-novelty period.** (A–B) Quantification of total distance traveled and mean velocity showed that animals of all groups spent similar amount of time moving within their home cages. (C) Representative examples of trajectories covered during the post-novelty period. Blue dots indicate start points, red dots indicate end points. Kruskal–Wallis, Dunn *post hoc* test, $p > 0.05$ corrected for the number of comparisons.

auditory cortex after thalamocortical reordering (*Pernia et al., 2017*). A similar pattern of IEG activation was also verified after visual deprivation (*Takahata & Kaas, 2016*).

The antibodies used have been previously validated (*Blanco et al., 2015*). The cross-modal up-regulation of the protein levels of egr-1 and c-fos detected in the present study is congruent with the known induction kinetics of these IEG (*Herdegen et al., 1991*; *Herdegen & Leah, 1998*; *Lonergan et al., 2010*), likely in association with the differential expression of NMDA receptors (*Chaudhuri, 1997*; *Kaczmarek & Chaudhuri, 1997*).

The present results add to the notion that cross-modality can be induced by the transient deprivation of stimulus from a sensory modality (visual, in this case), i.e., without any damage to the sensory periphery (*Pascual-Leone & Hamilton, 2001*; *Merabet et al., 2008*). However, since we only evaluated factors involved in a rapid response (one or three hours after visual deprivation), more studies are required to comprehensively characterize this phenomenon downstream of the initial genomic response.

Human brain imaging studies have shown that the visual cortex is significantly activated when subjects close their eyes and imagine objects (*Kosslyn et al., 1995*). Such visual mental imagery occurs when "a visual short-term memory (STM) representation is present but the stimulus is not actually being viewed" (*Kosslyn & Thompson, 2003*). In rodents, mental imagery can influence subsequent encoding and recognition processes of landmarks (*Karimpur & Hamburger, 2018*). In the present study we cannot rule out that our experimental design triggers imagery in the rats. However, since they explored the objects in the dark without any prior visual presentation, remains to be determined whether visual images could be formed for objects never encountered before.

In a previous study with the same experimental design, electrophysiological and molecular changes in V1 cortex were interpreted as nonspecific effects of arousal (*Ribeiro et al., 2007*). Assuming this hypothesis as correct, V1 cortex responses during whisked-based exploration of novel objects in the dark would represent only a general alert signal, carrying no specific information about object identity. However, in a subsequent study we analyzed with more depth the electrophysiological responses of individual V1 and S1 neurons of rats submitted to the same paradigm of exploration in the dark (*Vasconcelos et al., 2011*). This study revealed no significant difference in V1 and S1 neuronal responses to the explored objects, and thus reinforced the proposal that V1 cortex is in fact recruited during the whisker-based exploration of objects in the dark. Further studies evaluating IEG expression in S1 cortex using a control group with anesthetized whiskers shall help draw a more complete picture of the cellular changes in primary cortical areas whisker-based exploration in the dark.

In face of the present results, one key question concerning activation of visual areas by tactile stimulation is: why does it occur? Primary sensory areas have been shown to prefer a given modality but to quickly engage in cross-modal processing depending on task demands, a property named metamodality (*Pascual-Leone & Hamilton, 2001*). The functional recruitment of the visual cortex for tactile processing in healthy adults after a short period of visual deprivation is believed to reflect the unmasking of pre-existing connections (*Merabet et al., 2008*).

The visual cortex can be activated by locomotion (*Dipoppa et al., 2018*). To assess whether this effect could explain the results we evaluated the locomotion of the animals during the post-novelty period. We found that all groups spent similar amounts of time moving within their home cages, which rules out non-specific activation induced by locomotion.

## CONCLUSIONS

Our results show that even a short period of dark (3 h) is able to engage neurons in the V1 cortex in non-visual processing, leading to the up-regulation of molecular cascades that link sustained membrane depolarization with long-term gene regulatory changes. Further studies are required to a better understanding of the circuit dynamics associated with IEG activation following visual deprivation.

### Funding

This study was supported by grants from the Pew Latin-American Program in Biomedical Science, FINEP 01.06.1092.00, INCT-CNPq/MCT INCEMAQ 704134/2009, CNPq Universal 481506/2007-1, CAPES/SECYT 152/08. AASDAP–Brazil provided supplementary support. The funders had no role in study design, data collection and analysis, decision to publish, or preparation of the manuscript.

### Grant Disclosures

The following grant information was disclosed by the authors:
Pew Latin-American Program in Biomedical Science: FINEP 01.06.1092.00.
INCT-CNPq/MCT INCEMAQ: 704134/2009.
CNPq Universal: 481506/2007-1.
CAPES/SECYT: 152/08.

### Competing Interests

The authors declare there are no competing interests.

### Author Contributions

- Catia M. Pereira and Marco Aurelio M. Freire conceived and designed the experiments, performed the experiments, analyzed the data, prepared figures and/or tables, authored or reviewed drafts of the paper, approved the final draft.
- José R. Santos and Joanilson S. Guimarães performed the experiments, approved the final draft.
- Gabriella Dias-Florencio performed the experiments, analyzed the data, prepared figures and/or tables, approved the final draft.
- Sharlene Santos performed the experiments, prepared figures and/or tables, approved the final draft.

- Antonio Pereira contributed reagents/materials/analysis tools, authored or reviewed drafts of the paper, approved the final draft.
- Sidarta Ribeiro conceived and designed the experiments, analyzed the data, prepared figures and/or tables, authored or reviewed drafts of the paper, approved the final draft.

## Animal Ethics

The following information was supplied relating to ethical approvals (i.e., approving body and any reference numbers):

All experimental procedures were approved by the Institutional Local Ethics Committee for the Use of Animals (ID 05/2007), in accordance with the NIH Guidelines for the Care and Use of Laboratory Animals.

## Data Availability

The raw data are provided in a Supplemental File.

## Supplemental Information

Supplemental information for this article can be found online at http://dx.doi.org/10.7717/peerj.5678#supplemental-information.

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
