# Peer review of "Non-visual exploration of novel objects increases the levels of plasticity factors in the rat primary visual cortex"

_PeerJ, doi:10.7717/peerj.5678_

## Round 0.1 · original submission · Major Revisions

Two reviewers have now read the study and each has major concerns. R1 finds a few issues but is mainly concerned with lack of novelty and the authors’ claims. Whereas PeerJ does not require novelty in its rubric (PeerJ will publish direct replications) the paper must be set appropriately within the context of other studies. Before resubmission, please explain how this study adds to the literature in the context of each and every one of the studies and points that are listed by R1. If, in fact, the paper solely replicated previous findings, that’s fine, but then it’s claims must be in alignment.

R2’s analysis is much more concerning. S/he has major concerns about validity and methodological propriety of the analysis, and in fact reran the statistical analysis and found the authors to be wrong in their reporting of findings. That is unacceptable and the claims must be adjusted to indicate solely those findings that are supported by evidence.

Thus each and every one of both Reviewer’s concerns must be addressed (except for the issue of novelty, which is not a requirement at PeerJ) before PeerJ can accept the manuscript for publication.

·

Basic reporting

No Comment

Experimental design

See "General Comments"

Validity of the findings

See "General Comments"

Additional comments

There is little doubt that touch interacts with vision, even at the primary sensory level, as has been shown by numerous studies (Merabet, Lotfi B., et al. "Combined activation and deactivation of visual cortex during tactile sensory processing." Journal of neurophysiology 97.2 (2007): 1633-1641.; Merabet, Lotfi B., et al. "Rapid and reversible recruitment of early visual cortex for touch." PLoS one 3.8 (2008): e3046.; Shams, Ladan, and Robyn Kim. "Crossmodal influences on visual perception." Physics of life reviews 7.3 (2010): 269-284.; Vasconcelos, Nivaldo, et al. "Cross-modal responses in the primary visual cortex encode complex objects and correlate with tactile discrimination." Proceedings of the National Academy of Sciences 108.37 (2011): 15408-15413.; Lunghi, Claudia, and David Alais. "Touch interacts with vision during binocular rivalry with a tight orientation tuning." PLoS One 8.3 (2013): e58754.).

I am therefor not sure what the authors are trying to achieve with yet another study that demonstrates that the primary visual cortex responds to tactile stimulation. The interesting question is ignored! i.e. “WHY does touch trigger a response in V1?”

What the authors do not mention, neither in the introduction nor in the discussion, is that the results can be explained simply by visual imagery. Many neuroimaging studies of visual mental imagery have revealed activation of V1 (Kosslyn, Stephen M., and William L. Thompson. "When is early visual cortex activated during visual mental imagery?" Psychological bulletin 129.5 (2003): 723.). It is therefore plausible that tactile exploration of objects triggers visual imagery.

I guess that the authors, by exposing the rats to novel (and not familiar) objects, automatically did rule out visual imagery to explain the results; there is however no reason to believe that tactile exploration of unfamiliar objects in the dark does not trigger visual imagery.

I wish the authors had included a condition where rats explore familiar objects (for instance objects placed into the cage, days or weeks before the experiment). My prediction is that it would be easier for the rats to visualize those familiar objects, compared with novel objects (in the dark), and that the expressions of the molecular markers, for the “familiar” condition would even be stronger than for the “novel” condition. Any outcome, comparing a “familiar” with a “novel” condition would yield information that contributes to our understanding of why touch interacts with vision at the level of V1.

I encourage the authors to do one of the following:

- Either address all my concerns and provide (very) convincing arguments of why this study is novel and contributes to our understanding of cross-model interaction AND discuss the possibility of visual imagery (or why to exclude it as a possibility), in the introduction and/or conclusion.

- OR (preferred!!) perform additional experiments, adding a familiar objects condition, and change the scientific question from “does touch interacts with vision at the V1 level?” (already answered) to “why does touch interact with vision?” (interesting!).

·

Basic reporting

Previous work has shown that mRNA levels of certain immediate-early genes (IEGs) are elevated in V1 following whisker-based exploration of novel objects in a dark arena. Pereira et al. are essentially providing a replication study where they build on these previous results and show an elevation of the corresponding protein product. The manuscript is clearly written; however, the findings are somewhat over-sold: there is no evidence in the paper for long term plasticity or cortical reorganization, I would caution the authors to refrain from such speculation. The methods are described clearly. The two major issues (detailed below) relate to experimental design and a statistical error. Short of these being addressed the work is not suitable for publication. However, if the issues are remedied the paper represents a solid piece of histological work and could be considered for publication in PeerJ.

Minor points:
1. Third to last sentence in the abstract (line 49) was probably intended to read: “three hours after exploration” rather than “after three hours of exploration”
2. In the discussion, linking IEG expression to long term plasticity is a bit of a jump, the results certainly don’t warrant the phrase “indicate directly”

Experimental design

Major issues:
1. Locomotion (more precisely, all type of movement, including rearing, grooming, active whisking etc.) also activates the cortex, in fact it was recently shown to be one of the strongest activators, even comparable to relevant sensory stimulus (see preprints from Harris and Churchland labs). It is easy to imagine that animals in control boxes, with no foreign objects to explore, spend more time sitting quietly / asleep than their “stimulated” counterparts. Short of demonstrating that the animals spend similar amount of time moving (locomoting, grooming etc.) the results can be just as easily explained without cross-modal activation. Satisfactory demonstration could involve quantification of infra-red video footage or some other motion sensor and should not be particularly costly or time consuming.

Minor points:
3. It would be important to describe what environment are the animals exposed to in the 40 minutes / 2 hours and 40 minutes between exposure and euthanasia, as this could have a profound impact on IEG expression. Discussing the time dependence of IEG expression would also be helpful to allow the reader to contextualize the observations.
4. Specificity of an antibody should be determined in a mouse where the antigen was knocked out, using serum with no antibody only proves that the concept of immunolabelling works, not that the antibody labels its target. If the KO strains are not available, the minimum would be to cite previous work that validated these particular antibodies in some way.

Validity of the findings

Major Issue:

2. Since the authors provide all the data used in the quantification I have re-run their analysis using the exact same statistical method in the same software. If all data points are treated as independent observations there is still no significant difference between EGR-1 in control and at 3 hours. I am not sure why the authors claim a difference there. C-Fos and pCaMK2 calculations match the manuscript. However, this way of calculating differences is incorrect: the paper treats each image as an independent observation. Due to the inherently nested nature of the data (3 images collected from each animal, variance between animals is comparable to variance within each animal), results from the 3 sections / animal should be averaged and statistics should be done on a basis of 1 animal = 1 observation (n). Calculated correctly the only significant difference was between control and 1 hour after exposure for all three proteins. There is no detectable difference between either control and 3-hour or 1-hour and 3-hour time points.

Minor points:
5. Nuclear EGR-1 staining after exploration is clearly visible and the difference in pCaMK2 staining is appreciable as well on Figure 2 but it is hard to see the if there is any change in c-Fos staining. Is it just a poor example?
6. There are very visible luminance differences in the example images which might be problematic since the results hinge on accurate image quantification. The authors should explain why there are such large differences in the images if, as stated, they were processed all together.

---

## Round 0.2 · accepted · Accept

One reviewer was available to reread your manuscript and judged it ready to publish. Congratulations!

# ·

Basic reporting

In their revised manuscript the authors have addressed the concerns I brought up in the previous round. I recommend the publication of the manuscript in PeerJ.

Experimental design

-

Validity of the findings

-

Additional comments

-